# Using Principal Components Analysis and IDW Interpolation to Determine Spatial and Temporal Changes of Surface Water Quality of Xin’anjiang River in Huangshan, China

**DOI:** 10.3390/ijerph17082942

**Published:** 2020-04-24

**Authors:** Wenjie Yang, Yue Zhao, Dong Wang, Huihui Wu, Aijun Lin, Li He

**Affiliations:** 1College of Renewable Energy, North China Electric Power University, Beijing 102206, China; wjyang1109@126.com; 2Chinese Academy for Environmental Planning, Beijing 100012, China; zhaoyue@caep.org.cn (Y.Z.); wangdong@caep.org.cn (D.W.); 3Beijing City Environment Pollution Control and Resource Reuse Engineering Research Center, Beijing University of Chemical Technology, Beijing 100029, China; 2019400052@mail.buct.edn.cn (H.W.); environbiol@mail.buct.edu.cn (A.L.)

**Keywords:** Xin’anjiang River, principal component analysis (PCA), surface water quality, inverse distance weighted (IDW)

## Abstract

This study was aimed at assessing the spatial and temporal distribution of surface water quality variables of the Xin’anjiang River (Huangshan). For this purpose, 960 water samples were collected monthly along the Xin’anjiang River from 2008 to 2017. Twenty-four water quality indicators, according to the environmental quality standards for surface water (GB 3838-2002), were detected to evaluate the water quality of the Xin’anjiang River over the past 10 years. Principal component analysis (PCA) was used to comprehensively evaluate the water quality across eight monitoring stations and analyze the sources of water pollution. The results showed that all samples could be analyzed by three main components, which accounted for 87.24% of the total variance. PCA technology identified important water quality parameters and revealed that nutrient pollution and organic pollution are major latent factors which influence the water quality of Xin’anjiang River. It also showed that agricultural activities, erosion, domestic, and industrial discharges are fundamental causes of water pollution in the study area. It is of great significance for water quality safety management and pollution control of the Xin’anjiang River. Meanwhile, the inverse distance weighted (IDW) method was used to interpolate the PCA comprehensive score. Based on this, the temporal and spatial structure and changing characteristics of water quality in the Xin’anjiang River were analyzed. We found that the overall water quality of Xin’anjiang River (Huangshan) was stable from 2008 to 2017, but the pollution of the Pukou sampling point was of great concern. The results of IDW helped us to identify key areas requiring control in the Xin’anjiang River, which pointed the way for further delicacy management of the river. This study proved that the combination of PCA and IDW interpolation is an effective tool for determining surface water quality. It was of great significance for the control of water pollution in Xin’anjiang River and the reduction of eutrophication pressure in Thousand Island Lake.

## 1. Introduction

With the acceleration of China’s industrialization and urbanization, a large number of toxic and harmful pollutants have been discharged into surface water bodies, posing a direct and potentiallypersistent threat to their ecological environment and human health [1,2]. Contaminants in water can cause acute or chronic poisoning in humans through direct drinking or pose serious health risks to humans through sewage irrigation [3]. Every year, 190 million people fall sick due to water pollution, and 60,000 people die from diseases caused by water pollution in China [4]. In addition, water pollution can cause mental illness. It has been reported that people’s mental health can be improved in the absence of water pollution [5]. Therefore, it is urgent to evaluate the comprehensive water quality, understand the situation of water pollution, and identify the main pollution sources, to protect water resources and control water pollution [6,7].

With more attention to the quality of water environments, water quality assessment methods are also increasing. Currently, the most common methods used by scholars include the index evaluation method [8,9], fuzzy evaluation methods [10], grey evaluation method [11], and multivariate statistical method [12]. Among them, principal component analysis (PCA), a multivariate statistical method, is widely used to identify the relationship between the original indicator variables and transform them into independent principal components [13]. This method eliminated the correlation between evaluation indicators and greatly reduced the workload of indicator selection and calculation [14]. In recent years, PCA has been widely used in various environmental problems, including comprehensive assessment of temporal and spatial changes in surface water and groundwater quality [15,16], exploration of the leading sources of pollution in contaminated areas [17,18], and optimization of water quality observation network systems [19].

Inverse distance weighted (IDW) method, one of the most commonly used geostatistical and mathematical interpolation techniques, has been applied to predict the target parameters in the field of hydrology science [20]. It was developed for mapping and predicting spatial distribution maps, such as water quality parameters [21], methane flux [22], and rainfall intensity [23]. In this study, IDW method was used to interpolate the spatial distribution of water quality scores and identify key regulatory areas of the Xin’anjiang River (Huangshan).

The Xin’anjiang River eventually flows into Thousand Island Lake, with the Huangshan section accounting for more than 60% of the lake’s annual inflow. As the largest artificial freshwater lake in China, Thousand Island Lake has an extremely important strategic ecological position. It is an important water source in the Yangtze River Delta region and is one of the rare aquatic ecological areas in China. As the primary water source for Thousand Island Lake, the Xin’anjiang River is the most important ecological security barrier in eastern China. Its water safety has a great impact on human health and the ecosystem of Thousand Island Lake. Therefore, understanding the water quality status and pollution sources of the Xin’anjiang River is of great significance for effective management. However, information on the sources of pollution and water quality assessment of the Xin’anjiang River has rarely been discussed in previous studies.

This study was conducted as a preliminary survey on water contamination, through 24 basic water quality indexes of the water samples from the Xin’anjiang River (Huangshan), with the following objectives: (1) investigate the current status of water pollution in the river; (2) conduct PCA to identify the spatial and temporal changes of water quality and possible pollution sources; and (3) employ IDW interpolation to produce water quality distribution maps from 2008 to 2017. The results could be used to support water quality management, control pollution sources, and protect water resources in the Xin’anjiang River.

## 2. Study Area

The Xin’anjiang River originates from the Wugujian mountain, the junction of Anhui province and Jiangxi province (located at 117°38’–18°56’ E, 29°25’–30°16’ N) (Figure 1). The Xin’anjiang River runs through the Anhui and Zhejiang provinces, with an area of 11,850 square kilometers and a total length of 293 kilometers. In Anhui province, the main stream of the Xin’anjiang River is 242.3 kilometers long and covers an area of 6440 square kilometers. The Xin’anjiang River is located in the north sub-tropical region, with an average annual temperature of above 15 ℃ and rainfall of 900–1700 mm [24,25].

The Xin’anjiang River has a drainage area of 5830 square kilometers in Huangshan city, and flows through the Xiuning and Shexian counties from west to east, entering into Thousand Island Lake in Zhejiang province from the monitor station of Jiekou. Tourism is the main industrial structure of the Xin’anjiang River Basin. However, with the rapid development of tourism and an increasing number of tourists, the emission of pollutants has also increased. In addition, the non-point source problem caused by surface runoff cannot be ignored due to abundant water resources and the mountainous terrain in the Xin’anjiang River Basin. Since the implementation of the ecological compensation system, many polluting companies in the Xin’anjiang River Basin have been closed or relocated. With the effective control of industrial and urban domestic sewage, agricultural non-point source pollution, caused by agricultural production and rural life, has become the primary pollution source. Thus, pollution poses a threat to the stability of the water environment in the Xin’anjiang River Basin.

## 3. Analysis Procedures

### 3.1. Sampling

To accurately reflect the water quality of the Xin’anjiang River and determine the areas with serious pollution levels, eight representative state-controlled sections were selected, as shown in Figure 1. These monitoring stations mainly cover the main stream and primary tributaries of the river. Water samples were collected 50 cm below the water interface in the middle of the river on a sunny day. The 24 basic water quality parameters, according to the environmental quality standards for surface water (GB 3838-2002), were measured monthly from January 2008 to December 2017 in the Xin’anjiang River. Petroleum, volatile phenol, anionic surfactant (LAS), Se, sulfide, cyanide, permanganate index (CODMn), biological oxygen demand (BOD), ammonia nitrogen (NH_3_-N), Hg, Pb, chemical oxygen demand (COD), total nitrogen (TN), total phosphate (TP), Cu, Zn, fluoride, As, Cd, Cr(Ⅵ), and fecal coliforms (FC) were determined using the basic analytical methods according to the environmental quality standards for surface water (GB3838-2002)). Electrical conductivity (EC), dissolved oxygen (DO), and pH of the groundwater were determined in situ using a Professional Plus (Pro Plus) multiparameter instrument. Since petroleum, volatile phenol, LAS, Se, sulfide, and cyanide below the detection limit were meaningless for PCA (0.01 mg/L, 0.002 mg/L, 0.05 mg/L, 0.003 mg/L, 0.004 mg/L, and 0.002 mg/L), the remaining 18 parameters were selected for the following analysis.

### 3.2. Principal Component Analysis

It is a difficult and complicated process to know the water quality status of the whole river basin, determine the influencing factors of water quality, and improve the water environment quality of the river basin [26]. To provide a holistic vision of all the variables involved in the system, PCA was used on the Xin’anjiang River [27,28]. 

The PCA method is composed of five main operation steps, as follows:

(1) The original data matrix is listed:$$X={\left(x_{\mathrm{ij}}\right)}_{n*p}=\left[\begin{array}{ccc} x_{11} & \cdots & x_{1p}\\ \vdots & \vdots & \vdots \\ x_{n1} & \cdots & x_{np} \end{array}\right]$$
where $x_{\mathrm{ij}}$ in the matrix is the originally measured data, n represents the monitoring station, and p represents each water quality parameter.

(2) Standardize the original data with *Z*-score standardization formula to eliminate the impact of dimension (Equation (1)) [29].
(1)$$x_{\mathrm{ij}}{}^{*}=\left(x_{\mathrm{ij}}-\bar{x_{j}}\right)/s_{j},$$
where $x_{ij}{}^{*}$ is the standard variable, $\bar{x_{j}}$ is the average value for jth indicator, and $s_{j}$ is the standard deviation for jth indicator.

(3) Calculate the correlation coefficient matrix, R, with standardized data and determine the correlation between indicators (Equation (2)) [30].
(2)$$R={\left(r_{\mathrm{ij}}\right)}_{p*p}=\frac{1}{n-1}\overset{n}{\underset{t=1}{\sum }}x_{\mathrm{ti}}{}^{*}*x_{tj}{}^{*}\;\;\;\;\;\;\left(i,j=1,2,\cdots ,p\right)$$

(4) Calculate the eigenvalues and eigenvectors of the correlation coefficient matrix, R, to determine the number of principal components.

The eigenvalues of the correlation coefficient matrix, R, are represented by $\lambda_{i}\left(i=1,2\cdots n\right)$ and their eigenvectors are $u_{i}\left(u_{i}=u_{i1},\;u_{i2},\cdots u_{in}\right)\left(i=1,2\cdots n\right)$. The λ value corresponds to the variance of the principal component. And the value of variance is positively correlated with the contribution rate of the principal components. Further, the cumulated contribution rate of the first m principal components should be more than 80%, which means:$\;\overset{m}{\underset{j=1}{\sum }}\lambda_{j}/\overset{n}{\underset{j=1}{\sum }}\lambda_{j}\geq 0.80$ [31]. The principal component is represented by Equation (3).
(3)$$F_{i}=u_{i1}x_{1}^{*}+u_{i2}x_{2}^{*}+\cdots +u_{in}x_{n}^{*}\;\;\;\;\;\left(i=1,\;2,\cdots ,n\right),$$
where $x_{i}{}^{*}$ is the standardized indicator variable. $x_{i}{}^{*}=\left(x_{i}-\bar{x_{i}}\right)/s_{i}$.

(5) The obtained principal components are weighted and summed to obtain a comprehensive evaluation function, as shown in Equation (4).
(4)$$F=\frac{\lambda_{1}}{\lambda_{1}+\lambda_{2}+\ldots +\lambda_{n}}F_{1}+\frac{\lambda_{2}}{\lambda_{1}+\lambda_{2}+\ldots +\lambda_{n}}F_{2}+\ldots \frac{\lambda_{n}}{\lambda_{1}+\lambda_{2}+\ldots +\lambda_{n}}F_{n}$$

All mathematical and statistical calculations were performed using Microsoft Office Excel 2016 and SPSS 22.0 (IBM, Armonk, NY, USA). 

### 3.3. IDW Method

IDW interpolation is a common method of interpolation in spatial analysis. This method uses a linear-weighted combination set of sample points to determine cell values [32]. Greater weight will be assigned to the points that are closest to the target location.

The unknown value, Z(S_o_), at point S_o_ is calculated using the following formula:(5)$$Z\left(S_{o}\right)=\sum_{i=1}^{n}W_{i}Z\left(S_{i}\right),$$
where *n* is the monitoring station, Z(*S_i_*) is the value at the sampled locations *S_i_*, and *W_i_* represents the weight of *S_i_*, defined as:(6)$$W_{i}=\;\frac{\frac{1}{d_{i}^{k}}}{\left(\sum_{i=1}^{n}\frac{1}{d_{i}^{k}}\right)}\;\;\;\;\;\;\;\;\;i=1,\;2,\cdots ,\;n,$$
where *d_i_* is the horizontal distance between the interpolation points and the points observed, and k is the power of the distance.

All interpolation calculations were performed with ArcGIS 10.2 software (Esri, Redlands, CA, USA).

## 4. Results and Discussion

### 4.1. Principal Component Analysis

To study the impact of each water quality parameter on water quality and reduce the computation load, PCA was used to analyze the original monitoring data [33]. The objective of PCA was to extract the primary information representative of the typical characteristics of the water environment from a large amount of data and represent it as a new set of independent variables of the principal component [34]. PCA reduces the dimensionality of a multivariate data set to a small number of independent principal components. Each principal component contains all the variable information, thus reducing the omission of information [35].

In this study, PCA was conducted on 18 water quality indexes for eight monitoring points in the Xin’anjiang River. First, the applicability of PCA was tested by the Kaiser–Meyer–Olkin (KMO) and Barlett tests. These tests were used to verify the adequacy of the sample [36] and the independence of each variable [37], respectively. The calculated results were KMO = 0.79 (>0.5) and Barlett test value = 0 (<0.05), indicating that the data is suitable for PCA.

#### 4.1.1. Correlation Matrix

After nondimensionalizing the original monitoring data, a correlation coefficient matrix was obtained using SPSS 22.0 software (IBM, Armonk, NY, USA), as shown in Table 1. EC, CODMn, BOD, COD, TN, TP, and fluoride showed a strong positive correlation (r > 0.7). These water quality indexes were basically oxygen consumption indicators, which further indicated the overlap information of water quality indicators and the applicability of PCA [38]. A significant positive correlation was observed between Hg, Pb, Zn, and Cr(Ⅵ) (r = 0.88~0.94). The significant relationship between heavy metal ions may be related to the emissions of surrounding industrial point sources.

#### 4.1.2. Factor Loadings

The eigenvalues of each principal component are shown in Figure 2. The scree plot helps us to choose the principal components and understand the basic data structure. It was observed that the slope became noticeably flatter after the third component. The first three principal components were preserved, which explained 87.24% of the variance in the dataset.

Table 2 presents the factor loadings of these three factors for the 18 variables. The first principal component (PC1), which explained 49.54% of the total variance, contained large negative loadings on DO (−0.82) and positive loading on EC (0.95), CODMn (0.90), COD (0.97), TN (0.96), TP (0.94), and fluoride (0.92). The factor loadings of PC1 indicated that it mainly included oxygen-consuming pollutants, which may be related to influences from rural domestic wastewater, agricultural non-point source, and municipal point source discharge [39]. Previous research has shown that COD and NH_3_-N in the Xin’anjiang River were mainly derived from human sewage and agricultural wastewater [40]. The TN and TP in Xin’anjiang River mainly originated from surface runoff loss of nitrogen and phosphorus from tea plantations [41]. The results indicate that nutrient pollution and organic pollution are major latent factors which influence the water quality, and the impact of non-point source pollution in the Xin’anjiang River cannot be underestimated.

The second principal component (PC2), explaining 24.03% of the total variance, is strongly correlated with Hg (0.97), Pb (0.90), Zn (0.97), and Cr(Ⅵ) (0.96). Heavy metal pollution mainly arose from industrial point sources around the river and vehicle exhaust, which are generally discharged into the river without a surface runoff. The third principal component (PC3), explaining 13.67% of the total variance, is strongly correlated with the pH (0.89) of river water. The relationship between the pH value of a water body and other water quality indexes is complicated. Although the industries around the Xin’anjiang River have stopped production or have relocated, the impact of heavy metals on water sources is latent and long-lasting. Heavy metal pollution left over from industrial production should arouse the attention of relevant departments.

#### 4.1.3. Factor Scores

Factor scores were listed in Table 3. According to the results of Table 3, the monitoring points were divided into three groups, as shown in Figure 3.

The analysis of Figure 3 allows identification of groups that have taken similar values for certain analysis parameters. We defined a total of three groups that contributed to correlations between analysis parameters.

Group 1 showed negative correlation with PC1, which was distinguished by high DO values and low CODMn, COD, TN, TP, fluoride, Hg, Pb, Zn, and Cr(Ⅵ) concentrations. This group represents oxygen-rich types that are less contaminated by organic pollution, nitrogen, phosphorus, and heavy metals.

Group 2 showed positive correlation with PC1, which is characterized by relatively low DO and high CODMn, COD, TN, TP, and fluoride concentrations. This group represents the comprehensive organic pollution and nitrogen and phosphorus nutrients.

Group 3 showed strong positive correlation with PC2, which has a higher content of heavy metals such as Hg, Pb, Zn, and Cr(VI). The water quality of this type of water is significantly influenced by industrial activities on both sides of the Xin’anjiang River, which are largely related to human activities.

#### 4.1.4. Composite Score

The principal component scores were calculated with the variance contribution rate as the weight, and the composite score, F, was obtained afterward [42]. The comprehensive scores of monitoring stations in the Xin’anjiang River are shown in Table 3. 

The positive and negative scores of principal components do not represent the absolute water quality of the Xin’anjiang River, but represent its relative quality. The value of the comprehensive score, F, is negatively correlated with the river water quality; the smaller the value is, the better the water quality of the river [43]. In addition, the water quality index (WQI) method, which is a standard method to validate the results, was used [44], as shown in Table 3. It was observed that the evaluation results of the WQI method were consistent with those of the PCA method.

The PCA method uses the comprehensive score to represent the overall water quality, which overcomes the shortcomings of multi-index analysis. This study fills the research gap of comprehensive water quality assessment in Xin’anjiang River. According to the results of the comprehensive ranking, we found that poor water quality in the Pukou sampling point presents a worrying scenario for the region. This study screened out the highly polluted areas for the water quality management of Xin’anjiang River.

### 4.2. Temporal and Spatial Distribution of Water Quality

To obtain meaningful water quality information, the temporal and spatial distribution trends of water quality were predicted. According to the statistical calculation results from the monitoring stations, the water quality status of unmonitored areas and the spatial distribution of water quality were obtained through IDW interpolation. The IDW method interpolates the data from 2008 to 2017 and forms a spatial and temporal distribution map.

#### 4.2.1. IDW Method

IDW and Kriging are the most frequently used interpolation methods. IDW is simpler than Kriging, yet some studies showed that it surpassed the latter [45,46]. In addition, Kriging only works for normal distributions, while IDW has the ability to handle parameters that are not normally distributed [47]. The IDW method assumes that the values of the unsampled points are more similar to the values of the closer sampled points [48]. Since the change of water quality is continuous, and the water quality was greatly affected by closer observation points, IDW was used in this study.

#### 4.2.2. Spatial and Temporal Distribution Maps

This study first analyzes the evolutionary trend of water environments in the Xin’anjiang River over the past ten years. The spatial and temporal distribution maps were created by integrating the difference map from 2008 to 2017, as shown in Figure 4.

It can be concluded that the water quality of the Xin’anjiang River is generally stable, while that around the Pukou sampling point was inferior. Since 2008, water quality had no particularly noticeable improvement. As discussed above, organic and nutritional pollution and persistent heavy metal pollution are still very prominent.

## 5. Conclusions

In this study, PCA and IDW methods were used to determine the distribution of water quality in the Xin’anjiang River. This study first analyzes the evolutionary trend of comprehensive water environments in the Xin’anjiang River over the past ten years. The PCA method was used to extract the most significant indicator parameters affecting water quality and to identify the possible pollution sources of the Xin’anjiang River. The temporal and spatial distribution of water environment quality was mapped using the IDW method. The conclusions were as follows:

(1) Eighteen water quality indexes were reduced to three important principal components by PCA, explaining 87.24% of the total variance of the original data set. PC1 (49.54%) represented oxygen-consuming pollutants, indicating the influence of agricultural activities and domestic sewage on water quality. PC2 (24.03%) was contributed by heavy metals, which revealed the impact of human industrial activities. PC3 (13.67%) provided a positive correlation with the pH of the water sample.

(2) The spatial and temporal distribution map of water quality in Xin’anjiang River from 2008 to 2017 was made using the IDW method. The overall water quality is stable, while pollution management around the Pukou sampling point should be strengthened.

With the effective treatment of industrial point source pollution, the impact of agricultural and rural non-point sources on river water quality has gradually become prominent. However, heavy metal pollution left over from industrial production cannot be ignored. This study comprehensively analyzed the water quality of the Xin’anjiang River and identified the main factors affecting water quality and highly polluted areas. The results of this study could arouse more rational attention to drive the improvement of delicacy management for the ecology and environment of the Xin’anjiang River. Such an approach is recommended as a helpful tool for the sustainable management and development of river basins.

## Figures and Tables

**Figure 1 ijerph-17-02942-f001:**
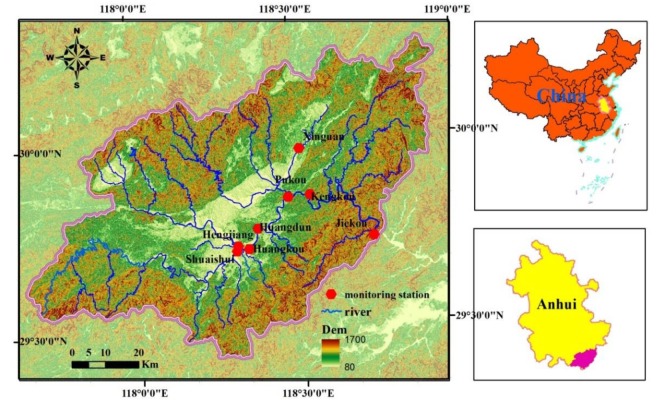
Sample sites distribution in the Xin’anjiang River (Huangshan).

**Figure 2 ijerph-17-02942-f002:**
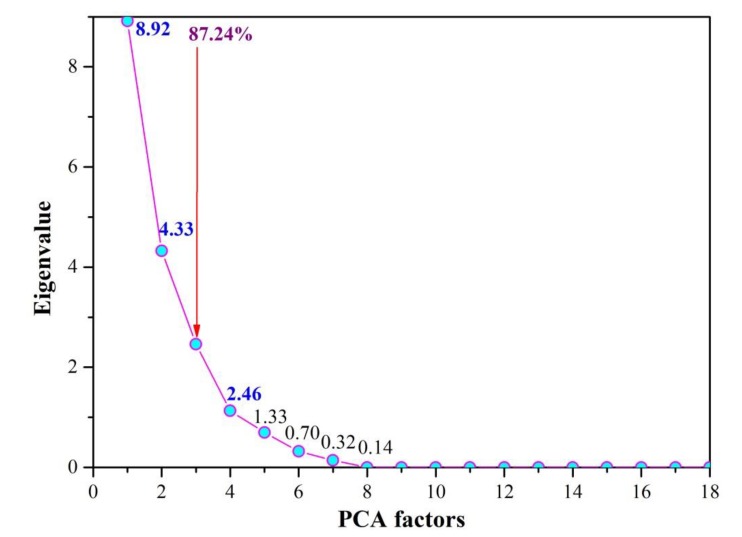
The scree plot.

**Figure 3 ijerph-17-02942-f003:**
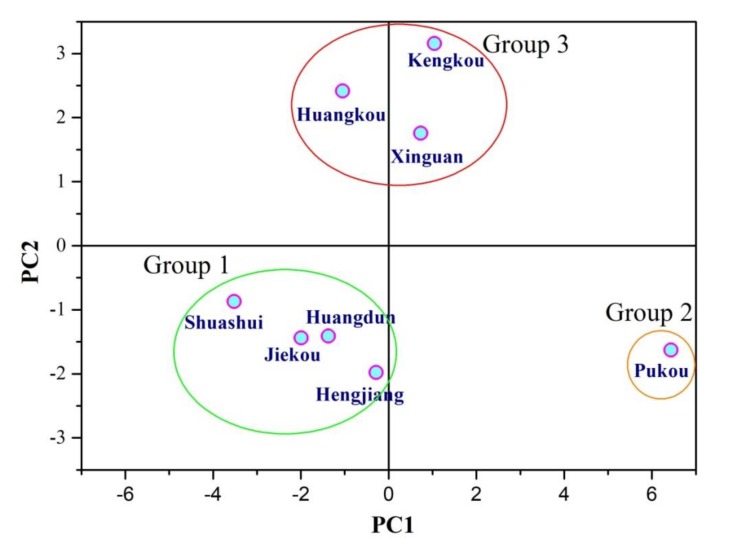
Factor scores (eight samples) of PC1 and PC2.

**Figure 4 ijerph-17-02942-f004:**
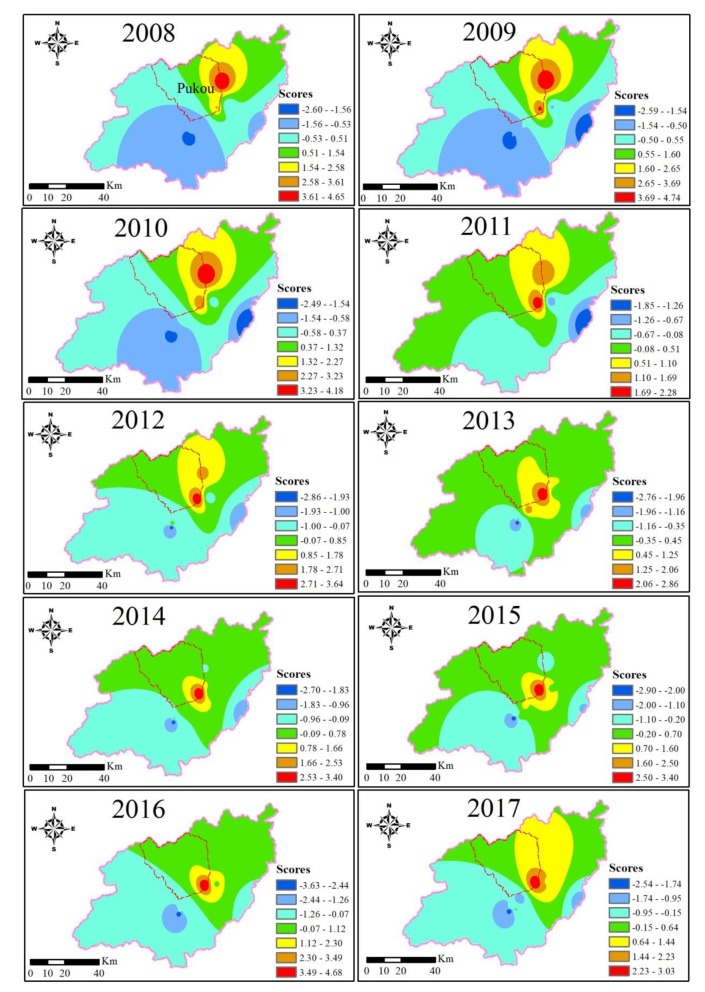
Spatial and temporal distribution of water quality in Xin’anjiang River.

**Table 1 ijerph-17-02942-t001:** Correlation coefficient matrix.

	pH	EC	DO	CODMn	BOD	Hg	NH_3_-N	Pb	COD	TN	TP	Cu	Zn	Fluoride	As	Cd	Cr(Ⅵ)	FC
pH	1.00																	
EC	−0.01	1.00																
DO	0.60	−0.65	1.00															
CODMn	−0.04	**0.91**	−0.58	1.00														
BOD	0.09	**0.78**	−0.20	0.70	1.00													
Hg	0.11	−0.12	−0.29	−0.15	−0.48	1.00												
NH_3_-N	−0.56	0.65	−0.74	0.68	0.59	0.03	1.00											
Pb	−0.06	0.28	−0.66	0.18	−0.23	**0.88**	0.28	1.00										
COD	−0.05	**0.96**	−0.68	**0.95**	0.73	−0.01	0.74	0.35	1.00									
TN	−0.30	**0.89**	−0.77	**0.95**	0.65	−0.09	**0.81**	0.29	0.95	1.00								
TP	−0.46	**0.82**	**−0.83**	**0.86**	0.61	−0.05	**0.89**	0.31	**0.88**	**0.97**	1.00							
Cu	0.29	0.79	−0.26	0.72	0.67	−0.14	0.38	0.11	0.70	0.57	0.43	1.00						
Zn	0.22	−0.03	−0.28	−0.19	−0.37	**0.94**	−0.04	**0.89**	0.02	−0.13	−0.10	−0.05	1.00					
fluoride	−0.14	**0.95**	−0.73	0.74	0.71	−0.05	0.66	0.38	**0.88**	**0.80**	0.77	0.67	0.11	1.00				
As	0.28	0.67	−0.38	0.36	0.42	0.08	0.11	0.40	0.58	0.38	0.28	0.44	0.34	0.77	1.00			
Cd	−0.37	0.70	**−0.89**	0.71	0.18	0.27	0.62	0.63	0.77	0.82	0.77	0.35	0.22	0.68	0.46	1.00		
Cr(Ⅵ)	0.27	−0.01	−0.25	−0.17	−0.35	**0.94**	−0.06	**0.88**	0.02	−0.13	−0.11	0.00	1.00	0.10	0.33	0.20	1.00	
FC	−0.43	0.25	−0.51	0.24	0.29	0.30	0.70	0.36	0.29	0.37	0.56	0.09	0.26	0.32	−0.16	0.16	0.25	1.00

Bold typeface: Strong correlation coefficient.

**Table 2 ijerph-17-02942-t002:** Loads of 18 variables in three principal components.

Eigenvalues	8.92	4.33	2.46
Cumulative (%)	49.54	73.57	87.24
Variable Factor	Factor 1	Factor 2	Factor 3
pH	−0.25	0.07	**0.89**
EC	**0.95**	−0.14	0.28
DO	**−0.82**	−0.32	0.36
CODMn	**0.90**	−0.25	0.12
BOD	0.67	−0.53	0.26
NH_3_-N	**0.83**	−0.06	−0.44
Hg	0.04	**0.97**	−0.03
Pb	0.43	**0.90**	−0.01
COD	**0.97**	−0.08	0.18
TN	**0.96**	−0.16	-0.10
TP	**0.94**	−0.11	−0.30
Cu	0.65	−0.19	0.50
Zn	0.07	**0.97**	0.17
fluoride	**0.92**	−0.01	0.21
As	0.54	0.20	0.62
Cd	**0.82**	0.26	−0.13
Cr(Ⅵ)	0.07	**0.96**	0.21
FC	0.44	0.24	−0.53

Bold typeface: Strong correlation coefficient.

**Table 3 ijerph-17-02942-t003:** Comprehensive evaluation value.

Monitoring Point	PC1	PC2	PC3	Composite Scores (Sorting)	WQI Values (Sorting)
Shuaishui	−3.5171	−0.8721	−2.0030	−2.5512 (1)	11.80 (1)
Hengjiang	−0.2822	−1.9778	1.8058	−0.4221 (4)	33.24 (5)
Hunagkou	−1.0502	2.4149	−0.8606	−0.0659 (5)	30.53 (4)
Dunhuang	−1.3722	−1.4120	−0.9634	−1.3191 (2)	23.30 (2)
Xinguan	0.7327	1.7571	2.2443	1.2518 (6)	35.37 (7)
Pukou	6.4341	−1.6283	−1.1362	3.0268 (8)	53.13 (8)
Kengkou	1.0445	3.1580	−0.4212	1.3970 (7)	35.23 (6)
Jiekou	−1.9896	−1.4396	1.3343	−1.3172 (3)	24.27 (3)

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
