# Peer review of "Using Principal Components Analysis and IDW Interpolation to Determine Spatial and Temporal Changes of Surface Water Quality of Xin’anjiang River in Huangshan, China"

_ijerph, 2020, doi:10.3390/ijerph17082942_

Round 1

Reviewer 1 Report

In this submission, authors have described analysis of water quality data obtained from river Xin’anjiang River in China. Data collected over the last 10 years is used for the analysis. I have following comments/questions:

Authors have provided some data analysis however the main hypothesis of the study is not clear. PCA is already an established method for water quality analysis, authors have also used Kriging method. But there is no explanation why Kriging is necessary here. Language is a major issue throughout the paper, which needs to be addressed. For example, lines say- “Since the first six indicators below the detection limit were meaningless for principal component analysis, the remaining 18 indicators were selected for analysis”. What is the detection limit? Why are these 18 parameters relevant for this study? Study area section also needs clear description of the frequency of measurements together with relevance of the selected sites. Some background on the sources of contamination is also necessary here. Line 123 mentions water quality index, but no definition or calculation method for WQI is provided. Xi in lines 126 and 127 needs to be clearly differentiated from X in matrix in line 121. Use of 7 references in the result and discussion section (4.1) is unexpected. Authors should use the materials and methods (Analysis Procedures in this paper) section to explain the process, where the references could be used to give relevance of the procedures used. Method to obtain Correlation Coefficient Matrix must be explain under Section 3. Section 4.2 requires more explanation. More details of the methods used – these must be covered under section 3. The basis of conclusion especially line 261 is not clear. What is the criterion for the validated?

Reviewer 2 Report

Dear Authors , this article can be improved using comparison of yours  obtained results with other researchers published sources eg " Evaluating the impacts of integrated pollution on water quality of the trans-boundary Neris (Viliya) River". Please write yours conclusions and recommendations according to yours obtained results and improve it following yours main objectives.

Reviewer 3 Report

In this manuscript, the authors applied a well-known multivariate data technique (principal component analysis) and spatial analysis (kriging) to evaluate the main water-quality variables that better represent water quality of the Xin'anjiang River (Huangshan, China) and their spatial and temporal distribution.

This study is on a topic of relevance and general interest to the readers of the journal. In my opinion, the manuscript is overall well organized. The strongest aspect of this work is represented by the experimental part. The dataset collected seem quite useful for the purpose of this study.

However, the manuscript lacks theoretical insights. This means that it is difficult for me to find a clear advance in knowledge and evident elements of novelty. Moreover, there are some scientific/methodological approaches that arose too many questions and do not make me feel comfortable to recommend this paper for publication simply with major revisions. In my opinion, this manuscript is at the limit of rejection. The Authors have to substantially improve their work.
In the following paragraph, there are comments meant to help the authors to improve their manuscript.

Major comments:
1. A significant concern that I have is represented by the lack of scientific questions. Which is the knowledge gap that the authors are filling with their work? Which are the possible implications of their findings? I strongly suggest that the authors should add an in-depth description of this information. There is a lack of this explanation in the abstract,
introduction and conclusions. Considering the current organization of the introduction, it is difficult to catch the knowledge gap that this study is trying to fill.
2. Another critical issue is represented by the lack of a clear explanation of the primary objective of this study in the abstract.
3. Another important scientific/methodological problem is represented by the use of kriging method. For kriging to work you have to conduct an intensive preliminary statistical analysis of the data known as "variography." The authors are not reporting in the manuscript any of these analyses. Furthermore, the number of data points used for
kriging method is very important. The authors are using only 8 data points (monitoring
stations), but it is usually recommended to have at least 15-20 data points to get satisfactory results. Based on these considerations, I believe kriging method is not the most appropriate one for this case study.
Please, consider the following publications:
1991, Myers,D.E., On Variogram Estimation in Proceedings of the First Inter. Conf. Stat. Comp., Cesme, Turkey 1987, A. Warrick and D.E. Myers, Optimization of Sampling Locations for Variogram Calculations Water Resources Research 23, 496-500
4. Please, revise the English style throughout the manuscript. It is not often appropriate for a scientific publication.

Minor comments:
5. The title should be re-phrased. “distribution of surface water quality” does not mean anything. Maybe “distribution of surface-water quality variables” or “distribution of surface-water quality indexes” can be better.
6. Line 14: the word “items” is not appropriate.
7. Lines 20-21: specify F1, F2, and F3.
8. The references has to be rewritten throughout the manuscript by following the
Instructions for Authors of the IJERPH journal.
9. Line 48: Re-phrase “to form”.
10. Line 62: Should “huangshan” be with capital letter?
11. Line 73: substitute “objectives” with “objective”.
12. Paragraph 2 should refer to Fig.1.
13. In paragraph 3.1, the authors should explain with more details hoe the monitoring campaign was carried out.
14. Fig.1: please, use the Inernational System Units (substitute miles with km2) and substitute “monitor station” with “monitoring stations”.
15. Paragraph 3.2: which software was used to run PCA?
16. Lines 116-118: the authors should better explain why PCA has been used. Please, refers to the following manuscripts: Gorgoglione, A.; Gioia, A.; Iacobellis, V. A Framework for Assessing Modeling Performance
and Effects of Rainfall-Catchment-Drainage Characteristics on Nutrient Urban Runoff in Poorly Gauged Watersheds. Sustainability 2019, 11, 4933.
Gorgoglione, A.; Bombardelli, F.A.; Pitton, B.J.L.; Oki, L.R.; Haver, D.L.; Young, T.M. Role of Sediments in Insecticide Runoff from Urban Surfaces: Analysis and Modeling. Int. J. Environ. Res. Public Health 2018, 15, 1464.

17. The paragraphs 3.2 and 3.3 has to be more balanced. The authors should decide if provide more information and formulas about kriging method, or remove the extra information about PCA.
18. Line 155: Substitute “In order to study” with “To study”.
19. Lines 155-156: Please, specify the sentence “reduce the amount of calculated data”. What does “calculated data” mean?
20. Line 157: “important information”…be more specific.
21. Line 159: “fewer new orthogonal factors”… be more specific.
22. Fig.2 vs. Table 2: It seems that in Fig. 2 the % of variance is not well represented. How the % can vary between 0 and 9? What the numbers in fig. 2 represent? The eigenvalues or the variance? The authors say that PC1 explains a variance of 33% (line 185). Why in Table 2 the cumulative variance is equal to 49.54%? Why in Table 2 only the positive strong correlation coefficients are in bold? Overall, Fig. 2 and Tab.2 present important results of this study, but they are not carefully depicted.
23. Line 199: substitute “loads” with “loadings”.
24. Fig. 3 does not add any information to the results represented in Fig. 2 and Tab.2; in fact, the authors do not provide any extra discussion. I suggest removing it.
25. Lines 218-219: “Calculate…score F”…rephrase this sentence, it is not clear.

26. Table 3: What does the composite score represent? Why calculating it?
27. Line 228: substitute “In order to obtain” with “To obtain”.

Round 2

Reviewer 1 Report

Thanks to authors for making corrections. However, all the corrections have not been made so some concerns still remain.

1- Language is still a major drawback in paper.

2- Line 109, still 'first 6' is mentioned - in the long list of parameters it doesn't make much sense to reader. Reason for this choice is also missing from the paper (although Authors indicated some reason in answer to my comments).

3- Lines 124 and 129 give different definitions for Xij, line 133 mentions xtj and xti. t is not defined. All of these equations require a clearer explanation.

4- F is the final water quality factor calculated in line 145. But, F is alo one of the water contamination parameters (is this Iron? then it me indicated as Fe).

5- Composite scores are calculated in Table 3, this should be supplemented with explanation on what these scores mean. How do these related to water quality? What does a negative score mean? Which score would relate to good water quality and which relates to poor water quality. Authors can use Water quality Index methods which are standard methods to validate their results. In absence of this comparison lines 254-256 are meaningless. 

6- Figure 1 and 4 are each drawn in a different way so it is not possible to make any visual comparison. Pukou area is mentioned as a problem area but it is not highlighted in either of the figures. Figure 1 indicates that only as a point (not an area). If these areas are mentioned in discussions, they need to be clearly highlighted.

Reviewer 3 Report

I appreciate that the Authors followed some of my suggestions to improve their manuscript. However, I still believe there is still an important methodological issue that does not make me feel comfortable to recommend this paper for publication.

In the enclosed file, I wrote in blue my second-round comments.

One more time, I suggest the Authors to substantially improve their work.

Round 3

Reviewer 1 Report

Dear Authors

Thanks for making some of the corrections. I still have following concerns

1- Pukou area is indicated in your analysis as a problem area since 2008 but it clear in Fig-4. You need to clearly highlight this area on the map.

2- You have replaced Kringing method with IDW and your results appear unchanged. There is also no explanation what prompted this change? The explanation about the IDW method  247-250 is not sufficient to justify its use. This article does not relate to a GIS system so this explanantion appears out of place.

3- Figure 3 plots the results that are in Table 3 but this is not mentioned anywhere. Table 3 appears on next page so this is not near the plot.

4- Page 3 lines 222-223 ' The water of this group is of nutrient rich type' does not make much sense.

5- Page 3- lines 225-226 what do mean by 'industrial activities on both sides'? which sides?

Reviewer 3 Report

I am glad that the Authors followed all my comments. I believe they substantially improved their manuscript and it is ready for publication.

Congratulations!
